# The Effect of Cued-Speech (CS) Perception on Auditory Processing in Typically Hearing (TH) Individuals Who Are Either Naïve or Experienced CS Producers

**DOI:** 10.3390/brainsci13071036

**Published:** 2023-07-07

**Authors:** Cora Jirschik Caron, Coriandre Vilain, Jean-Luc Schwartz, Clémence Bayard, Axelle Calcus, Jacqueline Leybaert, Cécile Colin

**Affiliations:** 1Center for Research Cognition and Neuroscience, Université Libre de Bruxelles, 1050 Bruxelles, Belgiumjacqueline.leybaert@ulb.be (J.L.); cecile.colin@ulb.be (C.C.); 2GIPSA-Lab, Université Grenoble Alpes, CNRS, Grenoble INP, 38402 Saint-Martin-d’Hères, Francejean-luc.schwartz@gipsa-lab.grenoble-inp.fr (J.-L.S.); clemence.bayard@gmail.com (C.B.)

**Keywords:** cued speech, multimodality, event-related potentials

## Abstract

Cued Speech (CS) is a communication system that uses manual gestures to facilitate lipreading. In this study, we investigated how CS information interacts with natural speech using Event-Related Potential (ERP) analyses in French-speaking, typically hearing adults (TH) who were either naïve or experienced CS producers. The audiovisual (AV) presentation of lipreading information elicited an amplitude attenuation of the entire N1 and P2 complex in both groups, accompanied by N1 latency facilitation in the group of CS producers. Adding CS gestures to lipread information increased the magnitude of effects observed at the N1 time window, but did not enhance P2 amplitude attenuation. Interestingly, presenting CS gestures without lipreading information yielded distinct response patterns depending on participants’ experience with the system. In the group of CS producers, AV perception of CS gestures facilitated the early stage of speech processing, while in the group of naïve participants, it elicited a latency delay at the P2 time window. These results suggest that, for experienced CS users, the perception of gestures facilitates early stages of speech processing, but when people are not familiar with the system, the perception of gestures impacts the efficiency of phonological decoding.

## 1. Introduction

The Cued Speech (CS) system is a visual mode of communication developed to facilitate access to spoken language for individuals with hearing impairments (HI). The CS code is composed of manual gestures that have two components: hand configuration, which is assigned to consonants, and position around the face, which is assigned to vowels. Each hand configuration and position around the face code a group of phonemes. Each phoneme in the group is visually contrastive and can be readily distinguished by lipreading. For instance, bilabial phonemes (/p/, /b/, and /m/) are coded by three different CS cues, and the manual cue which codes for the phoneme /p/ is also assigned for /d/ and /ʒ/, which is easily distinguished by lipreading. Importantly, when manual gestures are presented alone, speech perception remains ambiguous. However, when associated with lipreading information, a unique and precise phoneme can be perceived. Therefore, a CS cue is composed of two components: manual gestures (hand configuration and position around the face) and lipreading information [1]. In the French version, consonants are coded by eight hand configurations and vowels by five positions around the speaker’s face. The ability to combine CS manual gestures and lipreading information into a specific phoneme is acquired by implicit learning throughout the linguistic experience and consistent and repetitive exposition of the system [2,3]. Previous research suggests that children who are exposed to CS before the age of two, and who receive continued exposure to the system at home and at school, can learn to integrate CS cues with lipreading effortlessly. As a result, they develop accurate phonological representations of speech, which support their language acquisition and communication skills [3,4,5]. 

An intriguing research question in the domain of speech perception is how CS manual gestures interact with lipreading information in HI individuals. To address this question, Alegria and Lechat [3] investigated the effect of the congruent and incongruent presentation of CS gestures and lipreading information (without sound) on phoneme identification in HI children who were exposed to CS at home and at school. The rationale was to evaluate how CS gestures and lipreading information combine to form unitary percepts as a function of the weight attributed to each source of information. The results showed that when CS gestures were presented congruently with lipreading information, the accuracy of phoneme identification increased significantly. However, when the CS gestures were presented incongruently with lipreading information, children’s performance declined compared to the condition in which they relied solely on lipreading. Furthermore, the analysis of errors committed in incongruent conditions revealed that phoneme identification was not random, but rather reflected a compromise between lipreading and CS gesture information. For instance, when lipread information of the phoneme /v/ was being presented with an incongruent CS cue representing the phoneme group /p/, d/, and /ʒ/, participants consistently reported perceiving the phoneme /d/. This suggests that CS gesture has a significant weight on the final percept, but it is not the only source of information used by the children. Lipread information was likely used as an inference to constrain the available choices for the final percept. The authors considered it plausible that children’s perceptual system proceeded with the reasoning that “It’s likely to be a /d/ since if it were a /p/ or a /ʒ/, the lips would show it” [3]. Interestingly, the authors proposed that a similar interpretation can be used to understand the McGurk effect [6]. In the classical McGurk situation, participants systematically report perceiving /da/ when presented with incongruent auditory /ba/ and visual /ga/. This parallel can be explained by the participants taking lipread information into consideration as an inference of upcoming acoustic signals. The fact that participants’ response was more likely to be /da/ than /ba/ or /ga/ could be explained by the participants’ operating on the assumption that the bilabial viseme of /ba/ should be unambiguously seen by lipreading. This illusory percept indicates a compromise between lipreading and auditory perception, where the lipread information is matched against the auditory information that corresponds to a bilabial phoneme/ba/, and to solve the conflict, the perceptual system adopts a compromise between both sources of information. The compromise between lipread input and auditory perception in the McGurk effect has been taken as evidence that audio and visual information automatically interact in speech processing. Building on these findings, a follow-up study used the McGurk paradigm to investigate whether CS cues interact with auditory speech information in HI adults who were exposed to CS at young ages and were regular users of the system [7]. First, participants were presented with classical McGurk stimuli (auditory /pa/, lipreading /ka/) to verify whether or not they were sensitive to the expected effect (illusory perception of /ta/), similar to typically hearing individuals (TH). Second, CS gestures were associated with McGurk stimuli in three conditions: CS gestures congruent to lipreading information (i.e., /ka/), CS gestures congruent to the auditory input (i.e., /pa/), and CS gestures congruent to the expected illusional percept (i.e., /ta/), thus incongruent to auditory or to lipreading information. In the classical McGurk condition, participants’ responses were more likely to be a /ta/, indicating that HI individuals who are CS users are sensitive to the McGurk effect, similar to the TH group. However, when the CS manual gestures were congruent to either lipreading or the auditory input, the percentage of responses reporting /ta/ significantly decreased. In this condition, the perception of the CS gesture strengthened the incoherence between lips and sounds, and both labial and audio information were not bound. Contrastingly, when the manual gesture corresponded to the illusory percept (/ta/) and was incongruent with either lipreading or the auditory input, the percentage of responses reporting /ta/ was restored. Intriguingly, this result could suggest that the perception of the manual gesture increased the McGurk effect and facilitated the binding of audio and visual information. However, it is important to consider that CS manual gestures, being salient visual inputs, have the potential to attract or orient attention to the hands. Furthermore, for experienced CS users, manual gestures also convey phonological information that might influence phonetic decision-making. Therefore, based on the available data from this study, it is challenging to determine whether the observed increase in the proportion of responses reporting/ta/is primarily attributed to visual speech decoding or to an augmented McGurk effect. While this behavioral study [7] provides valuable insights, it does not definitively establish a direct interaction between the perception of CS gestures and audiovisual (AV) speech processing. Further research is necessary to investigate how the perception of CS gestures interacts with natural speech cues in AV speech processing. From a theoretical perspective, this research topic holds particular significance due to the limited number of studies that have specifically investigated the interaction between gestures conveying phonological information and speech processing [8]. While there is a growing body of literature on speech and gesture interaction, most studies have predominantly focused on semantic [9,10], iconic [11], or prosodic [12,13] gestures. Consequently, there is a noticeable gap in the literature on AV integration examining the effect of manual gestures conveying phonological information on automatic speech processing. From a clinical perspective, the interest in this research topic is driven by the fact that multimodal approaches in auditory rehabilitation hold great potential for compensating for challenges in speech perception, particularly in noisy listening conditions and for individuals with hearing impairments. By investigating the interaction between CS manual gestures and natural speech cues, we can gain insights into the potential role of this multimodal approach in auditory rehabilitation strategies. Understanding how CS perception influences auditory processing and the temporal dynamics of AV integration can inform the development of effective rehabilitation interventions.

In ecological conditions of speech perception, such as face-to-face communication, audio and visual information are related by a high level of cross-predictability due to their common underlying motor cause. Interestingly, the work from Kim and Davis [13] demonstrated that coupling a talker’s face with speech masked in noise enhances participants’ ability to detect the time interval during which a spoken sentence is presented, compared to when auditory speech was presented with a fixed face. Importantly, the authors also showed that the AV speech detection advantage was lost when the temporal correlation between auditory and video components was distorted. The observed AV advantage might be related to general (low-level) properties of AV perception of objects, and is not specific to speech. In multimodal speech perception, visual feature analysis of mouth articulatory movements helps to predict the content of auditory information, facilitating speech decoding. In an influential study, Klucharev et al. [14] manipulated the congruence between audio and video streams to differentiate neural responses that are speech-specific from general properties of AV perception. Participants were exposed to vowels (/a/, /o/, /i/, and /y/) in four conditions: auditory-only, visual-only, and two audiovisual conditions (congruent and incongruent). Results showed that the presentation of bimodal AV stimuli resulted in an amplitude attenuation of the first negative deflection (N1) of ERPs to bimodal conditions compared to unimodal conditions. Importantly, the observed amplitude suppression was present regardless of the congruence between auditory and visual stimuli. Additionally, the authors also observed a congruence-dependent modulatory effect at a later stage of speech processing. The presentation of phonetically congruent AV stimuli elicited an amplitude attenuation of the second positive deflection (P2) of the ERP responses. This latter modulatory effect, elicited by the presentation of phonetically congruent inputs, was interpreted as a neural marker of the fusion between visual and acoustic units into an AV percept. Subsequently, the study from van Wassenhove et al. [15] utilized an ERP paradigm in which participants were exposed to speech syllable (/pa/, /ta/, and /ka/) in unimodal (audio-only or visual-only) conditions and in bimodal AV conditions. The authors also manipulated the phonetic congruence between the audio and visual streams by creating McGurk stimuli (audio /pa/ and visual /ka/). The results from the bimodal AV conditions showed a reduction in the amplitudes of both the N1 and P2 ERP components compared to the unimodal conditions. Furthermore, the authors found that the components peaked earlier in the bimodal condition for phonetically congruent stimuli, but not for McGurk stimuli. Interestingly, at N1 and at P2 time windows, the magnitude of latency facilitation varied depending on the phoneme’s identity and the visual salience of its labial image (viseme); the latency facilitation was more pronounced for the bilabial /p/ phoneme, followed by /t/ and finally /k/. By capitalizing on the congruence-dependent latency facilitation and its modulation by visemic visual salience, the authors proposed an analysis-by-synthesis model of AV integration to explain their results within the theoretical framework of predictive coding. The proposed model suggests that bottom-up perceptual processing of visual speech information (pre-phonatory mouth movements) contributes to the creation of an online prediction model about the upcoming auditory signal. The amount and nature of visual information extracted during this period creates an abstract representation of the upcoming phoneme (internal predictor) that is continuously updated up to the point at which the auditory input is recorded. In phonetically congruent contexts, acoustic information matches against the internal predictor derived from visual inputs (visemes), and no prediction errors are computed. Furthermore, visemic input carrying information about the phoneme’s place of articulation would reduce spectral uncertainties in the auditory flow, constraining the auditory process to the second formant. Consequently, the computational costs of auditory processing would decrease, and the efficiency of speech processing would be maximized at the N1 and P2 time windows. These results paved the way for a series of studies on AV speech integration utilizing ERP analysis [16,17,18,19,20,21,22,23], which have tested the robustness of N1 and P2 modulatory effects. By employing a meta-analytical approach, the study of Baart et al. [24] confirmed the robustness of N1 and P2 amplitude attenuation and latency facilitation elicited by AV speech perception. Building upon these findings, the present study aims to investigate the effect of CS perception on the auditory processing of typically hearing participants by examining the temporal course of AV integration between CS gestures and natural speech cues through the use of event-related potentials (ERP) analysis. 

In CS production, manual gestures are performed slightly ahead of the mouth articulatory movement (≈200 ms), and both visual speech cues anticipate speech sounds [25]. Given the temporal coordination of CS perception, manual gestures likely provide a temporal cue of speech onset initiating speech processing. In the present study, the stimulus set consisted of three speech syllables (/pa/,/ta/,/ka/) presented in both unimodal and bimodal AV conditions. In the conditions involving CS gestures, participants observed one of three possible hand configurations (corresponding to/p/,/t/, or/k/), with the gestures placed in the same position around the face (corresponding to the vowel/a/). Therefore, the natural dynamics of CS perception and the spatial alignment of gestures and lipread information satisfies the spatial-temporal alignment prerequisites for AV integration [14,17]. 

We used an ERP paradigm to investigate whether CS perception modulates auditory processing in individuals who were either naïve towards the system or experienced CS producers. First, we sought to validate the experimental paradigm replicating findings of N1 and P2 amplitude attenuation and latency shortening elicited by the bimodal presentation of lipreading information and auditory information. Second, we aimed to verify whether the perception of CS gestures associated with lipreading information and the auditory input modulates the amplitude and latency of responses. At the N1 time window, we anticipated that the perception of CS gestures would provide a temporal cue of speech onset, preparing listeners for upcoming information and replicating previous findings that demonstrate amplitude attenuation and latency modulation. We expected these effects to occur in both groups thanks to the visual anticipation of CS gestures relative to speech onset. Likewise, at the P2 time window, we expected to observe amplitude attenuation and latency facilitation in both groups thanks to the effect of lipreading on speech decoding. Additionally, we predicted that modulatory effects occurring at the P2 latency range would be exclusively related to the presence of lipreading information in the group of naïve participants. Conversely, in the group of experienced CS producers, we anticipated that effects would be experience-dependent and related to the perception of CS gestures, which would facilitate speech decoding. Finally, our third goal was to verify whether CS gestures interacted with auditory speech processing when presented in isolation from lipreading information. In this condition, we expected to find N1 amplitude attenuation in both groups. These modulatory effects were expected to occur as a consequence of decreased uncertainty in the temporal domain and to be independent of the ability to decode CS gestures into speech information. At a later stage of speech processing, at which phonological decoding takes place, we expected to find different patterns of responses between the groups. Specifically, we expected P2 amplitude attenuation and latency facilitation to occur only in the group of people who are able to decode CS gestures into phonological information. In contrast, in the group of naïve participants, we did not anticipate finding any effects on either the amplitude or the latency of the P2 component. This prediction was based on the premise that CS producers possess the ability to mentally represent CS gestures in association with their corresponding phonological counterparts. When experienced CS producers are exposed to manual gestures, they are hypothesized to create an online prediction model of the upcoming phoneme until the lipreading information is observed. The simultaneous perception of lipreading information and manual gestures would provide perceivers with a strong prediction about the auditory input. Consequently, when the auditory input aligns with the internal prediction, it is anticipated that auditory speech processing will be facilitated. 

## 2. Materials and Methods

### 2.1. Participants

Thirty undergraduate students, native French speakers with no previous knowledge of CS (Mean Age = 27 years, SD = 3.45; 9 men, 21 women), and nineteen native French speakers who are experienced CS producers (Mean age = 31.7 years, SD = 6.21; 2 men, 17 women) participated in this research. All participants received monetary compensation for their participation in this study. The group of CS producers comprised individuals who had more than four years of regular practice, with seventeen of them working as professional CS interpreters, while two had learned to use CS at a young age (<5 years old) to communicate with a HI family member. The participants’ level of proficiency and age of acquisition were self-reported. All participants had typical hearing, as confirmed by a standard audiometric assessment (Interacoustics, Screening Audiometer AS608, Middelfart, Denmark) with hearing thresholds < 25 dB HL for pure tones ranging from 0.25 to 8 kHz. Moreover, they had normal or corrected-to-normal vision and no history of speech, language, cognitive, or psychological disorders. 

### 2.2. Stimuli Desing and Conditions

A native female French speaker and professional CS producer was videotaped while uttering and cueing CV syllables consisting of one of the /p/, /t/, and /k/ consonants articulated with the vowel /a/. To avoid stimulus predictability, six different exemplars of each syllable were chosen. The recordings were captured using a DV camera at a video sampling rate of 25 frames per second and at an audio sampling rate of 44.1 kHz. The recording was conducted in a controlled environment at the laboratory using consistent recording settings, placement, and luminance. To minimize differences in visual complexity in the stimuli, we extracted the soundtrack and two different masks (hand mask and lips mask) from each video. Subsequently, we embedded these different masks (with or without sound) on a fixed neutral image of the speaker to create experimental conditions (Figure 1). The video editing process was carried out using Adobe After Effects CC 2015 (v13.8) and Adobe Premiere CC 2015 software. Blue lipstick and colored dots on the hand and forehead of the speaker were used as reference points to align the different masks (hand and lips masks) on the image. A total of seven experimental conditions were created, including four unimodal conditions (audio-only or visual-only) and three bimodal AV conditions (as shown in Figure 2). In addition, catch trials were created for each condition by superimposing a small white point or adding an auditory bip to the video sound (also shown in Figure 2) [17]. Two exemplars of each syllable (/pa/,/ta/, and/ka/) were used for both audio and visual catch trials per condition. The auditory bip and small white dots were added approximately 40 ms before or after the acoustic syllable onset. 

### 2.3. Procedure

The experiment was conducted in a soundproof and electronically shielded room. The task was controlled using Presentation ^®^ software v.22.1 (www.neurobs.com, accessed on 6 July 2023). Instructions and videos (672 items, 630 stimuli trials, and 6 × 7 catch trial stimuli) were displayed on a screen at eye level and at approximately 70 cm from the participant’s head. The experiment consisted of six consecutive blocks of 112 trials (105 stimuli trials and 7 catch trials), and the total task duration was around 35 min. To avoid consecutive presentations of conditions of the same type, stimuli were pseudo-randomized, and all participants were exposed to the same stimuli order. The average duration of each trial was 1675 ms, including video fade-in/fade-out (300 ms), neutral fix image (variable frames), movement onset (variable frames), and sound onset (voice onset timing). The inter-trial interval (ISI) had a total duration of 1750 ms. Participants’ task was to press a keyboard bottom whenever they heard an auditory bip or saw the white dot (catch trials). Catch trials were presented in 6.25% of the total number of trials. To ensure that participants understood the instructions before starting the task, the instructor showed them an example of each catch trial, including one visual and one auditory example. Following this, participants completed a 5 min training session to familiarize themselves with the task before starting the main task. 

The target detection task (catch trials) was chosen instead of a phoneme discrimination task to fulfill the conditions of application of the additive model in our data analysis [17]. To be precise, the additive model requires that attentional and difficulty levels are equal across unimodal audio-only and AV conditions. As discussed by Besle et al. [26], phonological processing of lipreading information requires a higher visual attention level relative to processing syllables in bimodal AV conditions. Furthermore, asking participants to identify phonemes when CS gestures are presented may require higher effort and visual attention relative to conditions displaying natural visual speech cues such as lipreading. Thus, a phoneme discrimination task could lead to spurious effects in the computation of AV interaction that would prevent the use of the additive model and, thereby, data comparison across studies. 

### 2.4. Electroencephalographic (EEG) Recording

Continuous EEG data were recorded with BioSemi 64 channel-electrodes (10–20 system; Ag-AgCl active-electrodes, BioSemi, Amsterdam, The Netherlands) operating at a sampling rate of 256 Hz. Data were amplified using the BioSemi ActiveTwo AD-box EEG and referenced online to the Common-Mode-Sense/Driven-Right-Leg (CMS/DRL) reference electrodes (see www.biosemi.com, accessed on 6 July 2023). Electrode offsets relative to CMS/DRL were brought within 25μV before the beginning of the experiment. To minimize artifacts, participants were comfortably seated on an armchair.

### 2.5. Data Analysis

EEG data were imported to MATLAB (Math-Works Inc., Natick, MA, USA) using the EEGLAB toolbox [27]. EEG channels data were re-referenced offline to linked mastoids. Followingly, data were band-pass filtered using a zero-phase Hamming window finite impulse response (FIR) filter (low cut-off of 0.5 Hz and high cut-off of 40 Hz) with the pop_eegfiltnew() function of the EEGLAB toolbox. Data were segmented into epochs of 1000 ms (−500 ms to +500 ms), and the time zero (t0) corresponded to the sound onset of each syllable, which was individually determined by acoustical analyses using Praat [28]. Epochs with signal amplitude exceeding ±100 μV at any channel were automatically rejected to discard artefactual activity related to eye movements or muscular activities. Subsequently, to artifact rejection, epochs were baseline-corrected on a pre-stimulus interval of 300 ms (−300 ms to 0 ms). Following artifact rejection, an average of 10% of all trials were rejected in the group of naïve participants and 11% of all trials in the group of CS producers. 

The data pre-processing and analysis were conducted on a front-central cluster comprising six electrodes (F3, Fz, F4, C3, Cz, and C4). The choice of the front-central cluster of electrodes was based on classical studies which used electroencephalography to show neural correlates of AV integration in speech perception [15,17,18,21]. For all ERP components (N1 and P2), the variables of interest were the mean peak-to-peak amplitudes (μV) [22] and peak latencies (ms) with an average of activity taken over the six electrodes. Component time windows were defined based on the visual inspection of the grand average waveforms. Peak latencies were defined as the time point (ms) at which N1 and P2 components reached, respectively, the minimum and the maximum amplitude values. For each ERP component, amplitudes and latency values were extracted upon automatic peak detection using a MATLAB script. For each component, peak detection was performed within the following temporal windows: for P1, 70–130 ms post-stimulus onset (this value was used to calculate the N1 peak-to-peak amplitude); for N1, 80–180 ms; and for P2, 180–280 ms. Automatic peak detection was followed by careful visual inspection of individual data. The N1 peak-to-peak amplitude was calculated by subtracting the peak amplitude detected at P1 from the peak amplitude detected at the N1 time window. Similarly, the P2 peak-to-peak amplitude was calculated by subtracting the N1 peak amplitude from the P2 peak amplitude.

As in previous studies of AV integration [8,9,10,14], results were analyzed using the additive model. Accordingly, EEG signals obtained for bimodal AV stimuli presentation were compared to the algebraic sum of the EEG signals obtained for the presentation of the same stimuli in unimodal conditions (i.e., A + V ≠ AV). If the activity evoked by bimodal AV stimuli presentation is not significantly different from the sum of activities evoked by unimodal stimuli, it is assumed that audio and visual inputs were independently processed. On the other hand, if there is a significant difference, it is assumed that cross-modal interaction occurred, and differences emerged from a bimodal speech processing stage [15]. In the present study, we were interested in the following comparisons (Table 1): Audio.Lips compared to Audio + Lips; Audio.CS gesture compared to Audio + CS gesture; and Audio.Lips.CS gesture compared to Audio + Lips + CS gesture. In addition, we created an artificial condition composed of the sum of EEG signals obtained for the presentation of bimodal Audio.Lips stimuli and EEG signals were obtained for the presentation of CS cues-only condition. This latter condition (Audio.Lips + CS gesture) was artificially created to test whether effects (if any) were obtained in the Audio.Lips.CS gesture conditions are similar to those obtained for bimodal Audio.Lips stimuli. If ERP measures are similar, AV modulatory effects would be linked to lipreading and not to CS gesture perception. Conversely, if ERP measures are different, we assume that, in addition to the effect of the lips, there would be an additional effect linked to CS perception.

### 2.6. Statistical Analysis

Significant cross-modal interaction was assessed by the previously described planned comparisons of ERP measures between bimodal and the sum of unimodal conditions. Given that we were only interested in the comparison between related pairs (bimodal and the sum of the same signals presented in unimodal conditions) and not in every other possible comparison, paired *t*-tests were run to test differences on N1 and P2 mean peak latencies and peak-to-peak amplitudes between related pairs. We, therefore, considered significant cross-modal interaction comparisons yielding significant (*p*.adj < 0.05) difference after correction for multiple comparisons using the false discovery rate method for amplitude and latency separately.

## 3. Results

The results of this study are presented in four different subsections, each of which compares the neural activity evoked by bimodal stimuli presentation to that evoked by the sum of responses to unimodal stimuli presentation. Each subsection is divided into two parts, reporting results obtained in the group of naïve participants and in the group of experienced CS producers, respectively.

### 3.1. Audio.Lips Versus Audio + Lips

In this section, we aimed to test the cross-modal interaction between lipread information and the auditory input.

#### 3.1.1. Naïve Participants

Lipreading information associated with the auditory input had a significant effect on both N1 and P2 peak-to-peak amplitudes (N1: t = −4.03, df = 29, *p* < 0.001, *p*.adj < 0.001; P2: t = 3.63, df = 29, *p* = 0.001, *p*.adj < 0.05). Figure 3c,e, respectively, show that the mean peak-to-peak amplitude of N1 (M = −5.88, SD = 2.5) and P2 (M = 11.9, SD = 3.8) were significantly attenuated in Audio.Lips condition relative to the sum of unimodal signals Audio + Lips (N1: M = −7.15, SD = 2.4; P2: M = 13.5, SD = 4.1). No significant effect was found on components’ latencies (N1: t = 1.2, df = 29, *p* > 0.05, *p*.adj > 0.05; P2: t = 0.87, df = 29, *p* > 0.05, *p*.adj > 0.05). Figure 3a illustrates ERP profiles in both conditions and shows that responses are aligned in the temporal domain. Therefore, we replicate previous findings showing that AV presentation of lipreading information decreases the amplitude of N1-P2 responses. However, no facilitatory effects were found on the latency of either N1 or P2. 

#### 3.1.2. CS Producers

Similarly to the group of naïve participants, bimodal presentation of lipreading information had a significant effect on N1 and P2 peak-to-peak amplitudes (N1: t = 4.56, df = 18, *p* < 0.001, *p*.adj < 0.001; P2: t = −4.54, df = 18, *p* < 0.001, *p*.adj < 0.001). Figure 3d and Figure 3f, respectively, show that both N1 and P2 components were significantly attenuated in bimodal condition (N1: M = −6.9, SD = 3.07; P2: M = 12.5, SD = 3.07) relative to the sum of unimodal signals (N1: M = −8.75, SD = 3.78; P2: M = 14.8, SD = 5.7). Given that N1 latencies were not normally distributed, we conducted Wilcoxon paired test to compare conditions, and we found a significant N1 latency shortening effect (V = 97, df = 18, *p* 0.05, *p*.adj < 0.05): N1 latency was significantly shorter in Audio.Lips condition (M = 251.2, SD = 14.2) relative to that observed in the sum of unimodal signals (M = 256, SD = 13.7). For the P2 component, latencies were normally distributed, and Student’s *t*-test did not achieve a significance level (t = −1.16, df = 18, *p* > 0.05, *p*.adj > 0.05). Figure 3b illustrates ERP traces in both conditions. Note that the ERP trace representing Audio.Lips condition is slightly shifted leftwards relative to the Audio + Lips condition, which illustrates latency facilitation. Therefore, for this group, we replicated previous findings showing that lipreading information attenuated the amplitude of responses and speeded up speech processing at an early time range. 

### 3.2. Audio.Lips.CS Gesture Versus Audio + Lips + CS Gesture

In this section, we aimed to show how the perception of CS gestures interacts with lipreading information and auditory input. 

#### 3.2.1. Naïve Participants

Bimodal presentation of CS gestures associated with lipreading information and to the auditory input significantly modulated the amplitude of responses at N1 and P2 time ranges (N1: t = −5.54, df = 29, *p* < 0.001, *p*.adj < 0.001; P2: t = 5.25, df = 29, *p* < 0.001, *p*.adj < 0.001). Regarding components latencies, no effects were found neither at N1 time range nor at P2 time range (N1: t = −0.05, df = 29, *p* > 0.05, *p*.adj > 0.05; P2: t = 0.12, df = 29, *p* > 0.05, *p*.adj > 0.05). Figure 4a illustrates ERP traces in both conditions and shows that ERPs are aligned in the temporal domain. Figure 4c and Figure 4e, respectively, show that peak-to-peak amplitudes of both N1 and P2 were attenuated in bimodal condition (N1: M = −5.63, SD = 2.16; P2: M = 11.2, SD = 3.86) relative to the sum of unimodal signals (N1: M= −8.02, SD = 2.72; P2: M = 13.3, SD = 4.4). These results suggest that AV presentation of CS gestures associated with lipreading information decreased the amplitude of responses facilitating speech processing at early and late stages. Moreover, modulatory effects on the amplitude had no impact on the timing of responses, as no effects were observed in the latency domain. 

#### 3.2.2. CS Producers

We found a similar pattern of responses as in naïve participants. Both N1 and P2 mean peak-to-peak amplitudes were significantly different between conditions (N1: t = 5.04, df = 18, *p* < 0.001, *p*.adj < 0.001; P2: t = −4.29, df = 18, *p* < 0.001, *p*.adj < 0.001). In addition, we found a significant N1 latency difference between conditions (t = −4.37, df = 18, *p* < 0.001, *p*.adj < 0.05) that was not observed for the P2 component (t = −0.14, df = 18, *p* > 0.05, *p*.adj > 0.05). As shown in Figure 4d,f, components’ mean peak-to-peak amplitudes in bimodal Audio.Lips.CS gestures condition (N1: M = −6.95, SD = 3.51; P2: M = 12.12, SD = 4.84) were decreased relative to the sum of unimodal signals condition (N1: M = −9.76, SD = 3.75; P2: M = 15.1, SD = 6.04). Moreover, Figure 4b shows that the ERP trace from the bimodal condition was slightly shifted leftwards as N1 peak latency was shortened in this condition (N1: M = 158, SD = 8.77) relative to the sum of unimodal signals (N1: M = 170, SD = 10.6). 

### 3.3. Audio.Lips.CS Gesture Versus Audio.Lips + CS Gesture

In this section, we aimed to compare modulatory effects observed in Audio.Lips.CS gesture condition to the bimodal AV presentation of lipreading information. 

#### 3.3.1. Naïve Participants

As reported in the previous section, contingent AV presentation of lipreading information and CS gestures significantly modulated the amplitude of both N1 and P2 components. We conducted an additional ERP analysis to test whether the amplitude attenuation elicited by this later condition was similar to that observed in the AV lipreading condition. The mean peak-to-peak amplitude of N1 was significantly different between conditions. For the P2 components, the observed difference did not survive false discovery rate correction for multiple comparisons (N1: t = −2.7, df = 29, *p* < 0.05, *p*.adj < 0.05; P2: t = 2.13, df = 29, *p* < 0.05, *p*.adj > 0.05). As shown in Figure 5c, the mean peak-to-peak amplitude of N1 in bimodal Audio.Lips.CS gesture condition (N1: M = −5.63, SD = 2.16) was significantly decreased relative to the estimated sum of Audio.Lips + CS gesture signals (N1: M = −6.83, SD = 3.14). In addition, we observed a latency delay for the P2 component in Audio.Lips.CS gesture condition (M = 256, SD = 15) relative to the sum of Audio.Lips + CS gesture signals (M = 250, SD = 16). However, the observed difference did not survive false discovery rate correction for multiple comparisons (t = −2.52, df = 29, *p* < 0.05, *p*.adj > 0.05). In Figure 5a, note that ERP trace from bimodal Audio.Lips.CS gesture condition is slightly smaller at the N1 time window compared to Audio.Lips + CS gesture condition. Contrastingly, ERP traces are aligned at the P2 time window. These results suggest that the contingent presentation of CS gestures, lipreading information, and the auditory input further attenuated the peak-to-peak amplitude of the N1 component relative to AV lipreading presentation. Contrastingly, at the P2 time window, adding CS gestures to lipreading information and the auditory input did not elicit a robust modulatory effect. Taken together, these results suggest that the perception of CS gestures associated with lipreading facilitates the early stages of speech processing. 

#### 3.3.2. CS Producers

Similarly to the group of naïve participants, contingent AV presentation of lipreading information associated with CS gestures had a significant effect only on N1 peak-to-peak amplitude (t = 3.42, df = 18, *p* < 0.05, *p*.adj < 0.05). As shown in Figure 5d, N1 peak-to-peak amplitude was decreased in bimodal Audio.Lips.CS gesture condition (M = −6.95, SD = 3.51) relative to the sum of Audio.Lips + CS gesture condition (M = −7.85, SD = 3.04). Contrastingly, conditions had similar N1 latencies (t = 1.7, df = 18, *p* > 0.05, *p*.adj > 0.05,) (Figure 5b) and P2 peak-to-peak amplitude and latency (respectively, t = 0.83, df = 18, *p* > 0.05, *p*.adj > 0.05; t = 1.49, df = 18, *p* > 0.05, *p*.adj > 0.05) (Figure 5f). This result suggests that the perception of CS gestures in association with lipreading information and the auditory input significantly impacted the amplitude of responses at the N1 time range facilitating speech processing comparatively to bimodal AV presentation of lipreading information. 

### 3.4. Audio.CS Gesture Versus Audio + CS Gesture

#### 3.4.1. Naïve Participants

The association of CS manual gestures to the auditory input had no modulatory effects on the mean peak-to-peak amplitudes of any component (N1: t = −0.97, df = 29, *p* > 0.05; P2: t = 0.56, df = 29, *p* > 0.05). Figure 6c and Figure 6e, respectively, illustrate that there was no difference in mean peak-to-peak values of either N1 or P2. Moreover, conditions had similar N1 latencies (t = −0.89, df = 29, *p* > 0.05). For the P2 component, we found a significant effect on the latency domain (t = −3.1, df = 29, *p* < 0.001, *p*.adj < 0.05). As shown in Figure 6a, ERP traces from both conditions are aligned at the N1 time range, while at the P2 time range, responses are slightly delayed in bimodal Audio.CS gesture condition (M = 262, SD = 10.4) relative to the sum of the unimodal signals (M = 257, SD = 9.88). This result suggests that AV perception of CS gestures impacted the timing responses at later stages of speech processing delaying the latency of the P2 component. 

#### 3.4.2. CS Producers

As shown in Figure 6b, we have a different pattern of responses in the group of CS producers. The association of CS manual gestures to the auditory input modulated the peak-to-peak amplitude of the N1 component (N1: t = 2.5, df = 18, *p* < 0.05, *p*.adj < 0.05). Figure 6d shows that the mean peak-to-peak amplitude of N1 was slightly attenuated in bimodal Audio.CS gestures (M = −8.2, SD = 3.9) relative to the sum of unimodal signals (M = −9.1, SD = 3.6). Moreover, results showed that the latency of the N1 component was slightly shortened in bimodal Audio.CS gesture (M = 161, SD = 7.5) relative to the sum of unimodal signals (M = 167, SD = 10.7). However, the effect did not survive the false discovery rate for multiple comparisons (N1: t = −2.33, df = 18, *p* < 0.05, *p*.adj > 0.05). For the P2 component, student’s *t*-test did not achieve a significance level neither for the comparison between mean peak-to-peak amplitudes (t = −1.56, df = 18, *p* > 0.05) nor between mean peak latencies (t = 0, df = 18, *p* > 0.05). These results suggest that contingent presentation of CS gestures and auditory input modulated the amplitude of responses at an early stage but not at later stages of processing. Moreover, no robust effects were observed on the timing of responses. 

## 4. Discussion

The primary goal of the present study was to show how CS information interacts with AV speech cues and modulates speech processing in TH adults who were either experienced CS producers or naïve towards the system. We formulated hypotheses regarding how CS perception may interact at different stages of speech processing. Firstly, we hypothesized that CS gestures would provide a temporal cue for speech onset, thereby decreasing the uncertainty in the temporal uncertainty. As a result, we predicted that both experienced CS producers and CS-naïve participants would show attenuated responses at the N1 latency range, reflecting the influence of CS on early sensory processing, which we expected to be experience-independent. Additionally, we anticipated that the perception of CS gestures would elicit experience-dependent modulatory effects at the P2 time window, indicating its influence on later stages of speech processing. Specifically, we predicted that in the group of experienced CS producers, we would observe smaller and earlier P2 peaks, indicating that the information provided by CS gestures would facilitate speech decoding. Before delving into the results, it is important to acknowledge a potential limitation of the present study, namely the unequal number of participants in each group. Consequently, we were unable to test for statistically significant differences in the pattern of responses between the two groups. Therefore, we have focused our discussion on describing the results observed in each group, without drawing conclusions based on between-group differences. 

As a preliminary objective, we aimed to validate our ERP paradigm by replicating the findings of influential studies on audiovisual (AV) integration. These studies have provided compelling evidence that AV perception of lipreading information leads to attenuated auditory event-related potentials (ERPs) at N1 and P2 latency ranges [15,18,20,22]. Furthermore, earlier peak latencies of both N1 and P2 components have been reported in some studies [15,22], while others have shown earlier peaks, specifically in the N1 component [20,21]. In the initial phase of our study, our key findings align with the existing body of evidence, demonstrating that lipreading information interacts with auditory speech cues by attenuating the amplitude of responses at the N1 and P2 latency ranges. However, our results were less consistent with regard to component latencies, as we only replicated N1 latency facilitation uniquely in the group of CS producers and did not observe any effect on P2 latency. In this sense, our results appear to support findings from a previous work that suggested that N1 amplitude attenuation is more robust than N1 latency facilitation [20]. Moreover, they extend the discussion about the potential effects of experimental factors, such as task instruction and task type, on the pattern of responses at the behavioral level [29] and at the neurophysiological level [15]. One influential finding from van Wassenhove et al. [15] was that latency facilitation varied in function of the degree to which the visual signal predicts auditory targets, known as articulator-specific latency facilitation. In our study, participants were not specifically instructed to identify phonemes, but rather to detect audio and visual catch trials. Interestingly, Stekelenburg and Vroomen [20], who employed a similar task, did not observe N1 latency facilitation. These results suggest that specific task demands might potentially influence the timing of responses. Furthermore, another implication of this task choice is that, on data analysis, ERP data from all syllables are pooled together, and the averaging process could smooth differences in peak latencies undermining effects on components latencies. In light of the aforementioned, we consider that the less robust effect on the latency domain could be related to the type of task used and to our data processing. 

When CS gestures were associated with both the lipread information and the auditory input, the amplitudes of both N1 and P2 components were attenuated in both groups. In the group of experienced CS producers, N1 amplitude attenuation was accompanied by latency facilitation. Taken together, these results could suggest that adding CS gestures to natural AV speech cues elicited a similar pattern of responses as that observed in the AV lipreading condition. To confirm this hypothesis, we compared modulatory effects elicited by the AV presentation of lipreading information combined with CS gestures to those obtained to the estimated sum of responses obtained in the AV lipreading condition plus responses in the CS gestures condition. In both groups, the perception of CS gestures further attenuated the amplitude of the N1 component relative to bimodal AV lipreading perception. Given that the effect occurred independently of participants’ knowledge of CS, a plausible interpretation of the functional meaning of N1 amplitude attenuation could be that manual gestures provided a temporal cue of speech onset, thereby decreasing the uncertainty in the temporal domain. In this sense, these results could suggest that the perception of CS gestures prepares listeners for upcoming speech information, potentially decreasing the costs of the early stage of processing. However, these findings only partially confirm our hypothesis since we also predicted observing modulatory effects at the P2 time window in the group of CS producers. This result may suggest that having knowledge of CS is not enough to integrate it into an internal representation of speech phonological information, in line with the findings of one behavioral study [3]. Moreover, results from the fourth section extend this interpretation. In this section, we aimed to test the effect of bimodal AV presentation of CS gestures without lipreading on auditory speech processing. In the group of CS producers, results showed that bimodal presentation only modulated the amplitude of the N1 component and did not elicit modulatory effects at the P2 time window. In the group of naïve participants, the amplitude of responses was similar between conditions, but the latency of the P2 component was delayed when CS gestures were presented with the auditory input. These results were unexpected since we had anticipated observing N1 latency facilitation and amplitude attenuation in both groups as a consequence of the visual lead provided by gestures related to the sound onset [17]. Moreover, we were not expecting to observe modulatory effects occurring at the P2 time window in the group of naïve participants. These results indicate that the perception of manual gestures interacts with auditory speech processing in the former group, while in the latter group, it decreases the efficiency at which phonological information is processed. In individuals who are unfamiliar with CS, manual gestures may be too salient to ignore and may carry irrelevant information about the auditory input. Consequently, this could decrease the efficiency of phonological decoding. Conversely, for individuals who are familiar with CS, manual gestures are processed as visual speech cues that interact with the early stages of speech processing but are not fully integrated into multimodal speech perception. This finding would strengthen the hypothesis that AV interaction is not the same as AV integration as proposed by [30] “At early stages of speech processing, the early latency processes appear to cross-feed low-level information between the individual sensory cortices. This cross-feeding may modify the original input signal and can therefore be described as a multisensory interaction, but not necessarily as multisensory integration.” 

While this study successfully validated its aims by demonstrating how the perception of CS information interacts with auditory processing in TH individuals, there were some limitations that must be acknowledged. As mentioned above, the unequal group sample sizes prevented us from drawing firmer conclusions regarding the effect of intensive exposure to CS on AV speech processing. It remained unclear whether group differences could explain why we observed lipreading-induced N1 latency facilitation in the CS producer group but not in the naïve participant group. Moreover, it is worth noting that the age of exposure to CS has been found to be a significant factor influencing the ability to use CS information in speech decoding [4,5]. In our study, the majority of participants in the CS producer group (89%) were individuals who were exposed to CS during adulthood and were proficient in translating natural speech into CS code. However, their ability to decode CS information was limited to the identification of isolated phonemes, which met our inclusion criteria. Given the aforementioned considerations, further studies involving a comparable sample size of CS decoders relative to CS interpreters and naïve participants are needed to shed light on this issue. 

## 5. Conclusions

This study provided novel insights into how CS cues interact with AV speech processing in TH adults. Electrophysiological results showed that the perception of CS gestures, combined with lipreading information, accentuates speech processing facilitation at the N1 time window. It is worth noting that this effect was observed in both experienced CS producers and naïve participants. Contrastingly, when manual gestures are presented without lipreading information, we observed that the early stage of speech processing is facilitated in the group of CS producers. However, the facilitation observed at the early stage was not observed at a later stage, potentially suggesting that CS gestures might interact with AV speech cues but are not integrated into AV speech processing. The current study, therefore, suggests that, despite being artificial speech cues, CS information interacts with AV speech perception. However, the ability to use CS gestures as an internal representation that facilitates speech decoding may depend on early and intense exposure to the system. These findings pave the way for further research exploring the interaction between CS perception and AV speech processing in individuals who are exposed to the system in childhood and learn to decode gestures throughout their linguistic experience. This is generally the case for hearing-impaired individuals who are CS users. Understanding how CS cues modulate AV speech processing in individuals with hearing loss could provide valuable insights into the benefit of multimodal approaches following auditory rehabilitation.

## Figures and Tables

**Figure 1 brainsci-13-01036-f001:**
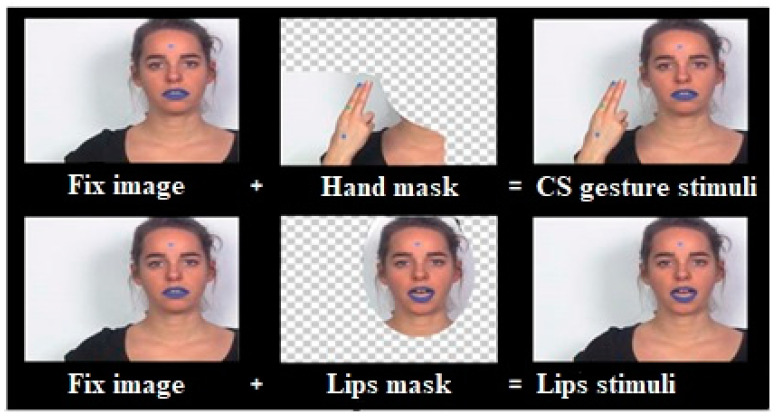
Illustration of the video editing procedure to create stimuli from two conditions. The hand mask extracted from the original video was embedded in the fixed neutral image to create a unimodal CS gesture condition (**top right**). To create unimodal Lips condition, a lips mask extracted from the original video was incrusted on the fixed neutral image (**bottom right**).

**Figure 2 brainsci-13-01036-f002:**
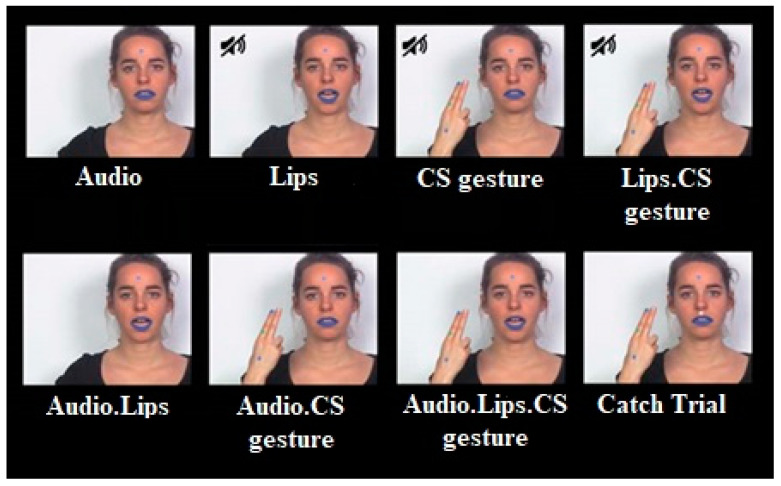
Illustration of the seven conditions of interest and of a catch trial. The catch trial illustrated here consisted of a white dot presented above the speaker’s lips (**bottom right**).

**Figure 3 brainsci-13-01036-f003:**
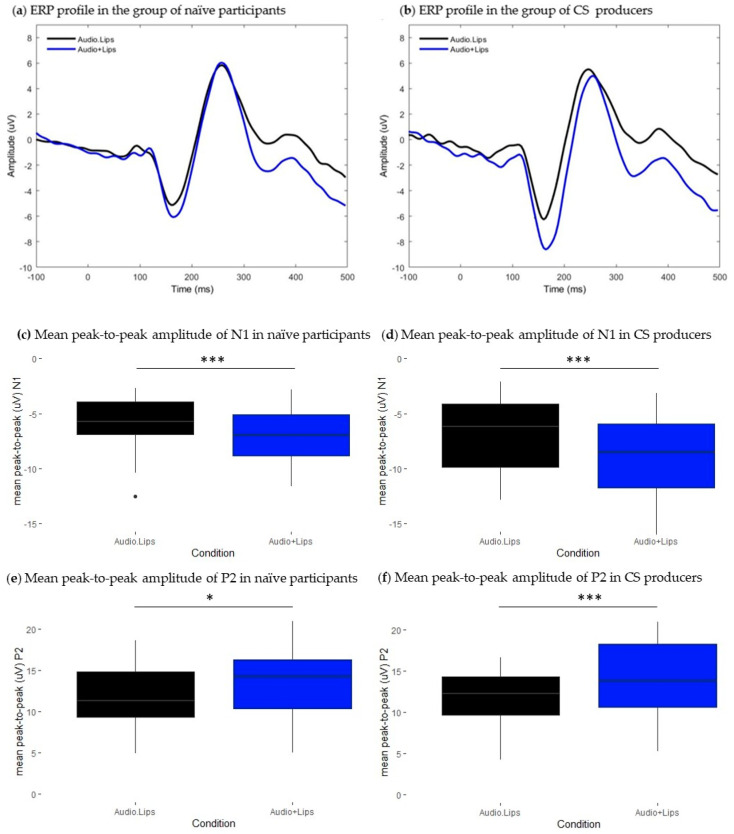
Results from the comparison between Audio.Lips and Audio + Lips conditions. (**a**,**b**): Grand-average waveforms extracted at the front-central cluster. Black traces represent ERPs of bimodal Audio.Lips condition, blue traces represent ERPs of the estimated sum of unimodal Audio+ Lips signals. (**a**) ERP profiles from the group of naïve participants and (**b**) ERP profiles from the group of CS producers. (**c**–**f**): Boxplots representing the distribution of mean peak-to-peak amplitude (uV) values of N1 and P2 ERP components. Black boxes illustrate values in the bimodal Audio.Lips conditions and blue boxes illustrate values in Audio + Lips conditions. Respectively, (**c**) and (**e**) show mean peak-to-peak values of N1 and P2 in the group of naïve participants, and (**d**) and (**f**), respectively, illustrate mean peak-to-peak values of N1 and P2 in the CS producers’ group. Error bars represent standard errors of the mean. * *p*.adj < 0.05; *** *p*.adj < 0.0001.

**Figure 4 brainsci-13-01036-f004:**
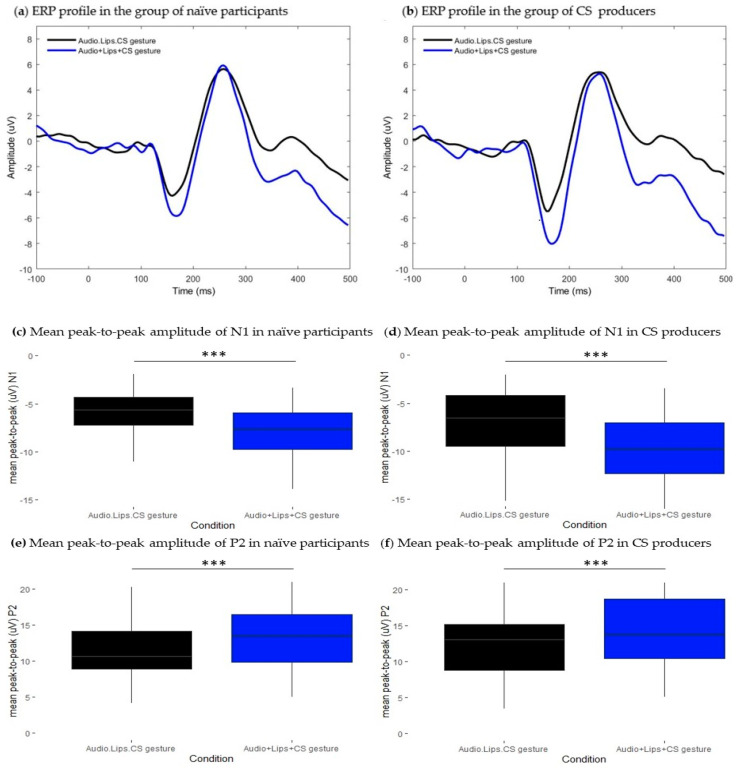
Results from the comparison between Audio.Lips.CS gesture and Audio + Lips + CS gesture conditions. (**a**,**b**): Grand-average waveforms extracted at the frontocentral cluster. Black traces represent ERPs of bimodal Audio.Lips.CS gesture condition, blue traces represent ERPs of the estimated sum of unimodal Audio + Lips + CS gesture signals. (**a**) ERP profiles from the group of naïve participants and (**b**) ERP profiles from the group of CS producers. (**c**–**f**): Boxplots representing the distribution of mean peak-to-peak amplitude (uV) values of N1 and P2 ERP components. Black boxes illustrate values in the bimodal Audio.Lips.CS gesture conditions and blue boxes illustrate values in Audio + Lips + CS gesture conditions. Respectively, (**c**) and (**e**) show mean peak-to-peak values of N1 and P2 in the group of naïve participants, and (**d**) and (**f**), respectively, illustrate mean peak-to-peak values of N1 and P2 in the CS producers’ group. Error bars represent standard errors of the mean. *** *p*.adj < 0.0001.

**Figure 5 brainsci-13-01036-f005:**
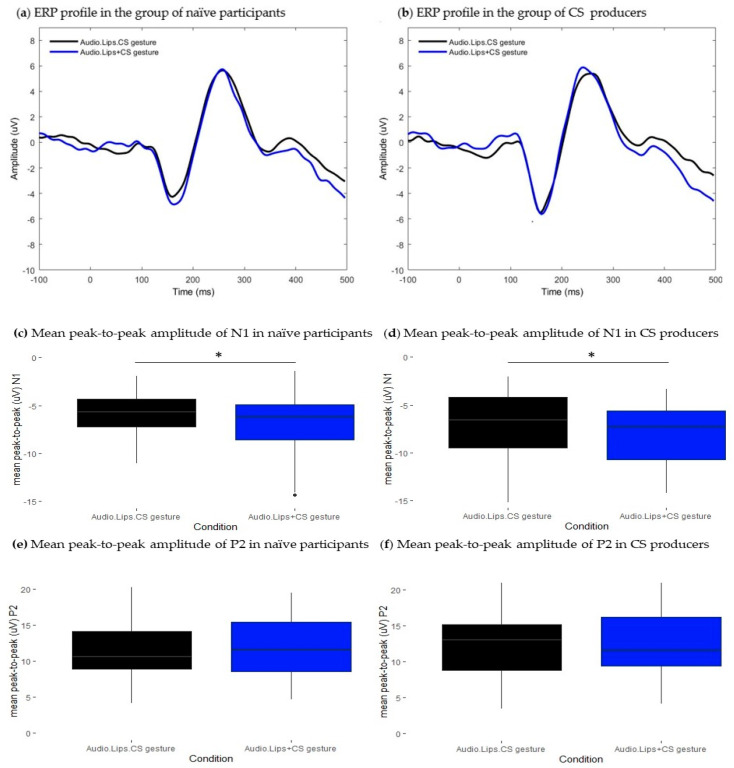
Results from the comparison between Audio.Lips.CS gesture and Audio.Lips + CS gesture conditions. Grand-average waveforms extracted at the frontocentral cluster. Black traces represent ERPs of bimodal Audio.Lips.CS gesture condition, blue traces represent ERPs of the estimated sum of bimodal Audio.Lips plus unimodal CS gesture signals (Audio.Lips + CS gesture). (**a**) ERP profiles from the group of naïve participants and (**b**) ERP profiles from the group of CS producers. (**c**–**f**): Boxplots representing the distribution of mean peak-to-peak amplitude (uV) values of N1 and P2 ERP components. Black boxes illustrate values in the bimodal Audio.Lips.CS gesture conditions and blue boxes illustrate values in Audio.Lips + CS gesture conditions. Respectively, (**c**) and (**e**) show mean peak-to-peak values of N1 and P2 in the group of naïve participants, and (**d**) and (**f**), respectively, illustrate mean peak-to-peak values of N1 and P2 in the CS producers’ group. Error bars represent standard errors of the mean. * *p*.adj < 0.05.

**Figure 6 brainsci-13-01036-f006:**
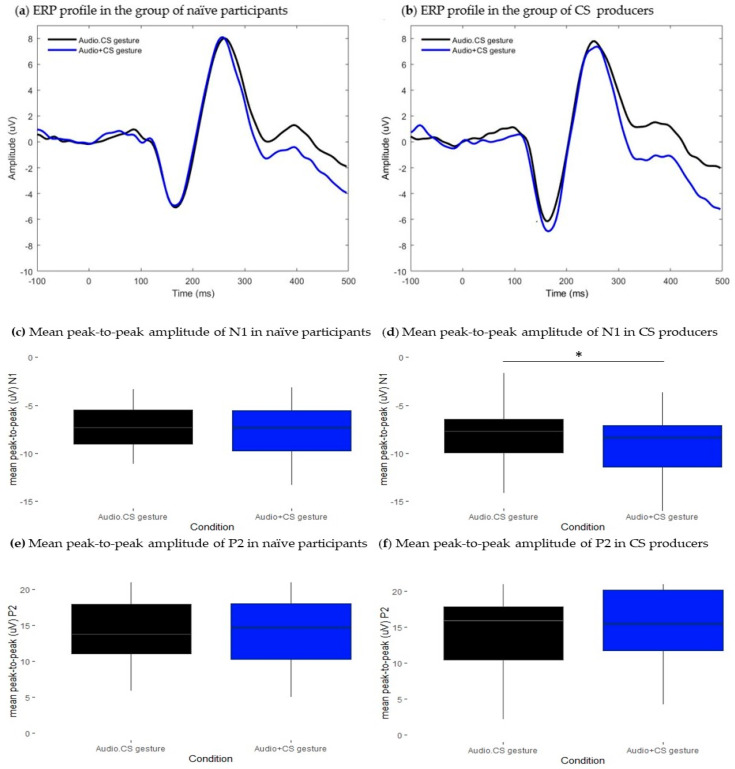
Results from the comparison between Audio.CS gesture and Audio + CS gesture conditions. Grand-average waveforms extracted at the frontocentral cluster. Black traces represent ERPs of bimodal Audio.CS gesture condition, blue traces represent ERPs of the estimated sum of unimodal Audio + CS gesture signals. (**a**) ERP profiles from the group of naïve participants and (**b**) ERP profiles from the group of CS producers. (**c**–**f**): Boxplots representing the distribution of mean peak-to-peak amplitude (uV) values of N1 and P2 ERP components. Black boxes illustrate values in the bimodal Audio.CS gesture conditions and blue boxes illustrate values in Audio + CS gesture conditions. Respectively, (**c**) and (**e**) show mean peak-to-peak values of N1 and P2 in the group of naïve participants, and (**d**) and (**f**), respectively, illustrate mean peak-to-peak values of N1 and P2 in the CS producers’ group. Error bars represent standard errors of the mean. * *p*.adj < 0.05.

**Table 1 brainsci-13-01036-t001:** Planned comparisons.

Planned Comparison	Bimodal Signal	Sum of Unimodal Signals
1	Audio.Lips	Audio + Lips
2	Audio.Lips.CS gesture	Audio + Lips + CS gesture
3	Audio.Lips.CS gesture	Audio.Lips + CS gesture
4	Audio.CS gesture	Audio + CS Gesture

List of the four families of planned comparisons. EEG activities of the bimodal AV condition (second column) were compared to the algebraic sum of EEG signals obtained in corresponding unimodal conditions (third column).

## Data Availability

The dataset will be made available from the corresponding author upon reasonable request.

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
