# Peer review of "The Effect of Cued-Speech (CS) Perception on Auditory Processing in Typically Hearing (TH) Individuals Who Are Either Naïve or Experienced CS Producers"

_brainsci, 2023, doi:10.3390/brainsci13071036_

Round 1
Reviewer 1 Report
Comments and Suggestions for Authors
This study was aimed at uncovering the extent to which Cued-Speech (CS) plays a role in integrated audio-visual processing of speech amongst naïve and experienced CS produces. While EEG was recorded Participants were presented with combinations of CS, audio speech, and video of speech, some integrated, some separate. Comparisons were made of auditory evoked potentials (N1 & P2) of the different conditions using the additive model to determine if the information in CS was integrated during auditory processing. The findings suggest CS may aid in speech processing for those with CS experience, while in naïve CS listeners they suggest CS may provide temporal cues that aid early speech processing while also hindering later speech processing. Future work will be needed to further understand the role CS provides in AV speech integration given the unexpected and somewhat inconsistent findings of the present study.
Overall, the study was well laid out, and the paper well written and concise. I have a few concerns that should be addressed before publication however. My biggest concern is there is no indication of any correction for repeated measures. Given that the analysis includes 32 t-tests, many of them using the same data just in different combinations, it is important to correct against the increased likelihood of spurious results due to the repeated measures. Please see the rest of my comments below
Data analysis 2.5 (pg 6) Clarification: Were any channels rejected/interpolated, or was the data cleaned in any other way (e.g. ICA to remove eye blinks)?
Data analysis 2.5 (pg 6-7): was peak detection done to find only local peaks, such that if the highest amplitude in the peak finding window was on one of the edges it would not be considered the peak?
Results section 3.4.1 (pg 12): it is stated no significant effects were found for mean peak to peak amplitudes, yet P<0.05 is reported for both N1 & P2s. The same is also found for N1 latencies in this section.
Suggestion for your plots: Please indicate significant comparisons where appropriate on the box plots. Was there a reason there are only box plots for the amplitude comparisons and none for the latency comparisons?
Reviewer 2 Report
Comments and Suggestions for Authors
This study examined the interaction between Cued Speech (CS), manual gestures facilitating lipreading, and natural speech using Event-Related-Potential (ERP) analyses. Both naive and experienced CS users with normal hearing were studied. Lipreading with CS gestures attenuated N1 and P2 responses, with CS producers showing facilitated N1 latency. Adding CS gestures increased effects in AV lipreading. CS gestures without lipreading produced similar responses in CS producers but caused P2 latency delay in naive participants, suggesting integration of CS gestures with AV speech cues and modulation of speech processing. Familiarity with CS influenced the efficiency of phonological decoding.
While the research design and findings are consistent with the literature on multisensory speech perception, the study can benefit from having a comprehensive literature review and integrated discussion on the theoretical development and research on the topic of multisensory speech perception. The novel contributions need to be highlighted and implications as well as future directions need to be specified. Some further clarifications in the method and results sections can also be helpful.
1. There is a large body of ERP work on audiovisual integration for speech perception. Visual speech perception refers to the ability to understand speech by observing the movements of a speaker's lips, facial expressions, and other visual cues. Some studies propose a functional differentiation between N1 and P2 in early sensory and perceptual processing of speech sounds (in terms of attentional processing and the involvement of bottom-up vs. top-down influences). This integration can lead to enhanced or reduced N1 and P2 responses depending on how the comparison conditions are set up (whether it is in comparison to audio only or whether the audiovisual information is congruent or incongruent). One idea is the role of visual attention. For instance, lip reading captures visual attention, directing focus towards the visual cues provided by the speaker's lips and facial movements. This increased attention to the visual stimuli can modulate N1 and P2 responses, potentially enhancing/facilitating early sensory processing of speech sounds. Another idea is the role of phonetic/phonemic processing. Lip reading can provide critical visual cues for perceiving phonetic and phonemic information. It helps in discriminating between different speech sounds, especially when the auditory signal is degraded or in noisy environments. The utilization of visual cues during phonetic and phonemic processing can influence N1 and P2 responses, reflecting the integration of auditory and visual speech information. Much of the interest in the research topic is driven by the fact that multisensory processing provides compensation in challenging listening conditions including speech perception in noise and hearing impairments. By relying on visual cues, individuals can compensate for the lack of auditory information. In such cases, N1 and P2 responses may be influenced by the increased reliance on lip reading and visual speech cues.
I encourage the authors to provide a thorough literature review to motivate the current research within the theoretical framework of predictive coding. Here are some references (not an exhaustive list). Relatively speaking, there is a lot more studies on influences of visual articulation information on auditory N1 and P2 responses than those on the use of hand gestures or cued speech.
Sun, J., Wang, Z., & Tian, X. (2021). Manual Gestures Modulate Early Neural Responses in Loudness Perception. Frontiers in neuroscience, 15, 634967. https://doi.org/10.3389/fnins.2021.634967
Mitterer, H., & Reinisch, E. (2017). Visual speech influences speech perception immediately but not automatically. Attention, Perception, & Psychophysics, 79, 660-678.
Talsma D. (2015). Predictive coding and multisensory integration: an attentional account of the multisensory mind. Frontiers in integrative neuroscience, 9, 19. https://doi.org/10.3389/fnint.2015.00019
Marin, A., Störmer, V. S., & Carver, L. J. (2021). Expectations about dynamic visual objects facilitates early sensory processing of congruent sounds. Cortex; a journal devoted to the study of the nervous system and behavior, 144, 198–211. https://doi.org/10.1016/j.cortex.2021.08.006
Stekelenburg, J. J., & Vroomen, J. (2007). Neural correlates of multisensory integration of ecologically valid audiovisual events. Journal of cognitive neuroscience, 19(12), 1964-1973.
Hisanaga, S., Sekiyama, K., Igasaki, T., & Murayama, N. (2016). Language/culture modulates brain and gaze processes in audiovisual speech perception. Scientific reports, 6(1), 35265.
Klucharev, V., Möttönen, R., & Sams, M. (2003). Electrophysiological indicators of phonetic and non-phonetic multisensory interactions during audiovisual speech perception. Cognitive Brain Research, 18(1), 65-75.
Irwin, J., Avery, T., Brancazio, L., Turcios, J., Ryherd, K., & Landi, N. (2018). Electrophysiological indices of audiovisual speech perception: Beyond the McGurk effect and speech in noise. Multisensory Research, 31(1-2), 39-56.
Huhn, Z., Szirtes, G., Lorincz, A., & Csépe, V. (2009). Perception based method for the investigation of audiovisual integration of speech. Neuroscience letters, 465(3), 204-209.
Ganesh, A. C., Berthommier, F., Vilain, C., Sato, M., & Schwartz, J. L. (2014). A possible neurophysiological correlate of audiovisual binding and unbinding in speech perception. Frontiers in psychology, 5, 1340.
Pinto, S., Tremblay, P., Basirat, A., & Sato, M. (2019). The impact of when, what and how predictions on auditory speech perception. Experimental Brain Research, 237, 3143-3153.
Molholm, S., Ritter, W., Murray, M. M., Javitt, D. C., Schroeder, C. E., & Foxe, J. J. (2002). Multisensory auditory–visual interactions during early sensory processing in humans: a high-density electrical mapping study. Cognitive brain research, 14(1), 115-128.
Stekelenburg, J. J., & Vroomen, J. (2012). Electrophysiological correlates of predictive coding of auditory location in the perception of natural audiovisual events. Frontiers in Integrative Neuroscience, 6, 26.
Kelly, S. D., Kravitz, C., & Hopkins, M. (2004). Neural correlates of bimodal speech and gesture comprehension. Brain and language, 89(1), 253-260.
Habets, B., Kita, S., Shao, Z., Özyurek, A., & Hagoort, P. (2011). The role of synchrony and ambiguity in speech–gesture integration during comprehension. Journal of cognitive neuroscience, 23(8), 1845-1854.
Pattamadilok, C., & Sato, M. (2022). How are visemes and graphemes integrated with speech sounds during spoken word recognition? ERP evidence for supra-additive responses during audiovisual compared to auditory speech processing. Brain and Language, 225, 105058.
Paris, T., Kim, J., & Davis, C. (2017). Visual form predictions facilitate auditory processing at the N1. Neuroscience, 343, 157-164.
2. The experimental setup included multiple conditions for comparison. It would be helpful to have a summary table for all the conditions to highlight the differences instead of just textual descriptions (Lines 284-294).
3. The quantification using peak-to-peak amplitudes for N1 and P2 needs further justifications. Did the previous ERP studies use this method? Why not use a peak window such as 40 ms, which seems to be more common in the ERP literature?
4. The statistical analysis using paired t-tests involved multiple comparisons, which requires correction method such as false discovery rate. As the comparisons involved between group and within group measures, it would seem necessary to use mixed effects models for each subsection by including the random intercept of inter-subject differences. The results and discussion need to specify exactly what each comparison reported in Sections 3.1, 3.2, 3.3 and 3.4 .was designed to address. Some of the results can be explained with the predictive coding framework nicely. For example, When manual gestures are presented without lipreading information, individuals who have knowledge of Cued Speech (CS) may experience facilitation in speech processing. This facilitation can be attributed to the predictive coding mechanisms in the brain. Through prior knowledge and experience with CS, individuals can generate predictions about the upcoming speech content based on the manual gestures they observe. These predictions can help the brain anticipate the corresponding auditory speech sounds, leading to enhanced processing and comprehension. By contrast, individuals without knowledge of CS may not benefit from the predictive coding mechanisms in the same way. Without prior expectations or predictions based on CS, their speech processing may not be facilitated when presented with manual gestures alone. This suggests that familiarity with CS is essential for predictive coding to effectively modulate speech processing in the absence of lipreading information.
5. Minor point: Please check and correct typos. For example, sited should be seated in Line 245.
Round 2
Reviewer 2 Report
Comments and Suggestions for Authors
I find the revision and responses satisfactory. I recommend acceptance.